# Intercultural Differences between Spain and Italy Regarding School Bullying, Gender, and Age

**DOI:** 10.3390/children10111762

**Published:** 2023-10-30

**Authors:** Antonio Ragusa, Ana Isabel Obregón-Cuesta, Emma Di Petrillo, Eduardo Maria Moscato, Jessica Fernández-Solana, Valeria Caggiano, Jerónimo J. González-Bernal

**Affiliations:** 1Rome Business School, Department of Education, 00196 Rome, Italy; ragusa@romebusinessschool.it (A.R.); dipetrillo@romebusinessschool.it (E.D.P.); 2Department of Mathematics and Computing, University of Burgos, 09001 Burgos, Spain; aiobregon@ubu.es; 3Department of Health Sciences, University of Burgos, 09001 Burgos, Spain; eduardomaria.moscato@posta.istruzione.it (E.M.M.); jejavier@ubu.es (J.J.G.-B.); 4Department of Education, University Roma TRE, 00154 Rome, Italy; valeria.caggiano@uniroma3.it

**Keywords:** bullying, Spain, Italy, primary education, secondary education, gender, age, categories

## Abstract

The objectives of this research were to establish the differences between Spain and Italy regarding the presence of bullying in primary and secondary schools, as well as to determine whether there are differences between experiencing or perpetrating bullying and gender and age in the practice of school bullying. To assess the EBIPQ scores in terms of country and gender, the chi-squared test was used, and ANOVA was applied for age. A total of 1536 students from primary and secondary schools in Spain and Italy participated in the study. Their ages ranged from 10 to 19 years (mean = 13.01, standard deviation = 2.19). The results revealed statistically significant differences in terms of bullying categories concerning the country of origin and gender, with a higher number of Italian participants in the role of “no victim aggress” and Spanish participants in the roles of “victim” and “victim and aggress”. Additionally, there were more boys in the role of “victim and aggress” and girls in the role of “no victim aggress”. Regarding age, statistically significant differences were found, with older students taking on the role of “aggress” on average, while younger students assumed the role of “victim”.

## 1. Introduction

Bullying can be described as a premeditated aggressive behavior that is repeatedly carried out with the intention of causing harm over an extended period of time. This behavior occurs in an environment where there is an imbalance of power, either perceived or real, which makes it difficult for the victim to protect themselves from the aggressor [1,2,3,4]. This definition allows us to distinguish bullying from occasional fights between young people who have similar conditions of physical, psychological, or social strength. Bullying involves deliberate and repetitive aggression with a clear power imbalance [5]. Sporadic fights between young people are characterized by isolated confrontations without a systematic pattern of abuse or intimidation [6]. The distinction between both concepts is crucial to properly identify and address harassment issues in the school environment [7,8,9].

Currently, bullying has gained significant global attention, showing a high prevalence in all nations and becoming a public health challenge [2,10,11]. This phenomenon has led to a considerable increase in research focused on the topic in recent decades [7]. In fact, the World Health Organization (WHO) considers it one of the primary health issues affecting childhood and adolescence [12]. The growing concern surrounding this issue has resulted in a broader recognition of its negative impact on society and has driven the search for effective solutions to address this serious social problem [4].

It affects children worldwide, with estimates of 24 million children and young people being victims of bullying and mistreatment in European schools each year [13,14]. Figures for Italy in both 2021 and 2022 showed approximately 32,600 cases, while Spain had 69,554 severe bullying cases during the same years [13,14,15]. A study reports that around 20–25% of young people are directly involved in bullying as victims, aggressors, or both [16]. The average prevalence is estimated to be 35% for traditional bullying and 15% for cyberbullying [4,17]. Although prevalence may vary significantly depending on the sample, age, methodology, etc., the existence of this significant problem and its profound consequences on the current and future mental health of students are undeniable [15]. 

Indeed, until a few years ago, research on school bullying has primarily focused on analyzing the “aggress”/”victim” dyad, overlooking the role of bystanders or “no victim-aggress” individuals in these episodes. This omission is due to the complexity of the bystander’s role, as it involves numerous and conflicting factors that influence their behavior in a context of intimidation [18,19].

Despite its complexity, it is essential to recognize the significance of the “no victim aggress” role in the dynamics of school bullying, as their behavior can have a significant impact on the perpetuation or prevention of these situations. Therefore, future research and intervention strategies must address the active involvement of “no victim aggress” in the bullying phenomenon to comprehensively tackle this serious issue. As a result, several roles have been identified, including the aggressor, the “victim”, the “victim and aggress”, and, finally, the role of the “no victim aggress” [18,20,21,22]. It is important to note that these roles are not static, as the same individual may assume different roles over time. Thus, to effectively address the problem of school bullying, adopting a multidimensional approach that recognizes the complexity of interactions between the different roles involved is essential [18,21]. 

Various studies have shown a high prevalence of bullying throughout the school years, with rates that tend to increase during early childhood and then gradually decrease towards the end of adolescence [17]. Moreover, changes in educational environments, such as the transition from primary to secondary school, can impact the frequency and manifestation of school bullying [18,20]. Social dynamics change, friend groups reconfigure, and interactions among students can become more complex [23]. Indeed, it is important to note that, although rates of school bullying may decrease in adolescence, the emotional and psychological impact of bullying experiences during childhood and adolescence can be long-lasting [24]. Therefore, it is crucial to continue working on the prevention and intervention of school bullying at all educational stages to ensure a safe and healthy school environment for all students. Creating a supportive and respectful atmosphere can contribute to the overall well-being and academic success of students, promoting positive social interactions and reducing the negative effects of bullying [7,10].

On the other hand, differences in the rates of school bullying based on gender have been observed. Research has shown that boys tend to experience higher rates of bullying compared to girls. This difference could be related to the distinct ways in which bullying is manifested between genders, where boys are often more involved in direct and physical behaviors, while girls may resort to subtler and emotional tactics [4,7,25].

Thus, the aims of this study were to establish the differences between Spain and Italy regarding the presence of bullying in primary and secondary schools. Additionally, the aim was to determine if there is a relationship between experiencing or perpetrating bullying, gender based on the country of origin, and age in the practice of school bullying.

## 2. Materials and Methods

### 2.1. Participants

The study included a total of 974 Italian students, with 513 students in secondary education distributed across the first (n = 162), second (n = 186), third (n = 187), and fourth (n = 165) grades. Additionally, 461 students were enrolled in high school, spread across the first (n = 101), second (n = 85), third (n = 125), fourth (n = 84), and fifth (n = 66) grades. In terms of gender, the sample consisted of 398 male students and 576 female students. These data were collected from three different public schools located in the city of Gela, within the province of Caltanissetta.

Furthermore, there was a sample of 562 students in Compulsory Primary Education (EPO) and Compulsory Secondary Education (ESO). Within this subgroup, 284 were male and 278 were female. The EPO students (n = 334) were enrolled in the fifth (n = 228) and sixth (n = 186) grades, while the ESO students (n = 148) were in the first (n = 134) and second (n = 94) grades. The sample selection was carried out using a cluster sampling method and was drawn from five different schools, including both public (n = 4) and private (n = 1) educational institutions located in the Autonomous Community of Castilla y León.

### 2.2. Instrument

The European Bullying Intervention Project Questionnaire (EBIPQ) is a tool designed to identify the prevalence of involvement in school bullying situations, including individuals who take on the roles of aggressors, victims, or both [26,27]. This questionnaire has demonstrated strong psychometric properties in multiple European countries, including Spain [8,10]. Each of the questionnaire’s subscales comprises seven items and is structured to assess the frequency of aggression or victimization, with these items encompassing various aspects of school bullying [12,28]. The initial seven items assess experiences of victimization, while the subsequent seven items pertain to involvement in aggressive behaviors. Students are asked to indicate how often they have engaged in or encountered each of the described scenarios over the past two months [29]. 

To identify the different participation roles, we followed the criteria defined by the scales. To determine the “victim” role, individuals who received ratings equal to or greater than 2 (once a month) in any of the items related to victimization were considered, along with scores equal to or less than 1 (one or two times) in all aggression items. Involvement in the “aggressor” role was calculated by considering individuals who obtained scores equal to or greater than 2 (once a month) in any of the aggression items and scores equal to or less than 1 (one or two times) in all victimization items. The “victim and aggress” roles were identified through scores equal to or greater than 2 (once a month) in at least one of the aggression and victimization items. Lastly, for the “no victim aggress” role, scores were lower than 1 (one or two times) in at least one of the aggression and victimization items [27,30].

The internal consistency of the instrument for the current sample collected for this study is 0.852. Additionally, the internal consistency analysis for the first 7 victimization items is 0.796 and, for the following 7 items, it is 0.804. Furthermore, the internal consistency of the scale for the samples used is 0.852 for the Spanish sample and 0.850 for the Italian sample.

The frequency is assessed over the preceding two months and is rated on a Likert-type scale ranging from 1 to 5, with the following response options: No; Yes, once or twice; Yes, once or twice a month; Yes, about once a week; and Yes, more than once a week [12,27]. In Appendix A, the EBIPQ questionnaire, in both Spanish and Italian versions, used for this study is presented.

### 2.3. Procedure

The study commenced with the initiation of contact with the school principals at the research site, where we communicated the research objectives. After obtaining their agreement to participate, the parents or legal guardians of the students in various classrooms provided informed consent for their children’s involvement in the study. Data collection involved the use of scales, and these scales were administered anonymously. The confidentiality of the collected information was rigorously maintained, with a clear understanding that it was solely intended for research purposes. Data collection took place during regular school hours, and detailed instructions were provided to ensure accurate completion. The questionnaires were individually filled out in an appropriate school setting, free from distractions, and the entire process adhered to ethical guidelines in accordance with the standards set by the American Psychological Association.

### 2.4. Statistical Analysis

The study commenced with an initial univariate analysis aimed at collecting descriptive information about the sample. Subsequently, several bivariate analyses were conducted to compare EBIPQ scores across gender and country, employing the Chi-square test. To investigate differences among various bullying-related categories derived from the EBIPQ scale and age, a one-way ANOVA was performed, followed by subsequent post hoc testing. The statistical significance level was established at *p* < 0.05, and all analyses were carried out using SPSS software version 25 (IBM Inc., Chicago, IL, USA).

## 3. Results

The sample consisted of a total of 562 Spanish participants, representing 36.6% of the sample, and 974 Italian participants, comprising 63.4% of the total sample. In total, 44.5% were male participants (n = 683) and 55.5% were female participants (n = 853). Likewise, the ages ranged from 10 to 19 years, with a mean age of 13.01 ± 2.19. 

In the Spanish sample, there were specifically 284 male participants (50.5%) and 278 female participants (49.5%), with an average age of 11.66 ± 1.20. In the Italian sample, there were 399 male participants (41%) and 575 female participants (59%), with an average age of 13.79 ± 2.25.

### 3.1. Association between Bullying Categories, Country, and Gender

Statistically significant differences are observed in the categorization of subjects into types of bullying and the country of origin of the sample (χ2 = 40.684; *p* < 0.001). 

Analyzing the corrected residual values, there are differences between the observed and expected frequencies for Spanish students in the categories of “no victim aggress”, “victim”, and “victim and agress”. However, there are no significant differences for Spanish students in the category of agress. Similarly, in the Italian sample, significant differences are observed for the same categories.

There are more Italian students (72.7%) in the “no victim aggress” category than Spanish students (57.3%), and there are more Spanish students in the “victim” (24.7%) and “victim and agress” (14.4%) roles than Italian students (14.5%; 9.5%).

Similarly, it can be observed in the categories with significant results that there are fewer Spanish students (322) in the “no victim aggress” category than expected (376.9) and a higher number (139) in the “victim” category (102.4) and “victim and aggress” (81) than expected (63.7). In contrast, in the Italian sample, there is a higher number (708) in the “no victim aggress” category than expected (653.1) and a lower number for “victim” and “victim and aggress” (Table 1).

Statistically significant differences are observed in the categorization of subjects into types of bullying and the combined category of gender + country of origin (χ^2^ = 60.476; *p* < 0.001).

Analyzing the corrected residual values, differences are evident between observed and expected frequencies for Spanish boys in the categories of “no victim aggress”, “victim”, and “victim and aggress”. However, no significant differences are found for Spanish boys in the “aggress” categories. For girls, significant differences exist only in the “victim” category but not in the other categories.

In Spain, among boys, 52.1% fall into the “no victim aggress” category, followed by 24.3% in “victim”, 18.7% in “victim and aggress”, and 4.9% in “aggress”. For girls, the highest percentage is in the “no victim aggress” category at 62.2%, followed by “victim” (25.2%), “victim and aggress” (10.1%), and “aggress” (9.4%). Notably, significant results in Spanish boys are found in the “no victim aggress” category, with a lower number (148) than expected (190.4), “victim”, with a higher number (69) than expected (51.8), and “victim and aggress”, with a higher number (53) than expected (32.2). Similarly, in girls, significant results are observed in the “victim” category, with a higher number (70) than expected (50.7).

Analyzing the corrected residual values, differences are observed between observed and expected frequencies for Italian boys in the categories “no victim aggress” and “victim”. However, no differences are found for Italian boys in the “aggress” and “victim and aggress” categories. As for Italian girls, significant results are obtained for the categories “no victim aggress” and “victim and aggress” but not for “victim” or “aggress”.

In Italy, 73.4% of boys fall into the “no victim aggress” category, followed by “victim” at 11.8%, “victim and aggress” at 11.5%, and, finally, “aggress” at 3.3%. The distribution among girls is similar, with the “no victim aggress” category at 72.2%, followed by “victim” (16.3%), “victim and aggress” (8.2%), and “aggress” (3.3%). Furthermore, in the categories with significant results for Italian boys, there are more in the “no victim aggress” category (293) than expected (267.6) and fewer in the “victim” category (47) than expected (72.7). Similarly, for girls, significant results are observed in the “victim” category, with more (415) than expected (385.6), and the “victim and aggress” category, with fewer (47) than expected (65.1) (Table 2).

### 3.2. Association between Bullying Categories and Age

There is a significant difference *p* < 0.001 between the ages of students in different bullying categories (F(3,1532) = 8.151, *p* < 0.001).

Students belonging to the victim category are those with the lowest average age, 12.54 years (SD = 1.88), followed by “no victim aggress” at 13.05 years (SD = 2.22), “victim and aggress” at 13.25 years (SD = 2.27), and, finally, those with the highest average age belong to the “aggress” category at 13.92 years (SD = 2.20) (Table 3).

A post hoc test was conducted to compare the means pairwise for each of the bullying categories.

Significant differences were observed between students with the “no victim aggress” category and students in the “victim” (*p* = 0.001) and “aggress” (*p* = 0.005) categories. Furthermore, significant differences were also observed between the “victim” category and students in the “aggress” (*p* < 0.001) and “victim and aggress” (*p* = 0.001) categories. Finally, those in the “aggress” category were significantly different from those belonging to the “victim and aggress” category (*p* = 0.049).

## 4. Discussion

The phenomenon of school bullying is a pervasive issue worldwide, and large-scale studies have been conducted in various countries, revealing a wide range of prevalence rates, spanning from a minimum of 10% to a maximum of 70% [3], despite the existence of antibullying protocols in nearly all regions.

In Spain, existing action protocols, such as those implemented by the Spanish Association for the Prevention of School Bullying (AEPAE), are primarily geared toward immediate victim protection, based on daily experiences with victims and their families. These protocols come into play once a bullying case has already occurred, and their nature is advisory, deliberate, and bureaucratic, focusing on documenting the case and delineating responsibilities. There is a proposal to extend the role of action protocols to include preventive measures, emphasizing raising awareness to reduce the incidence of bullying [31]. Furthermore, Spain addresses bullying through various national and regional laws, regulations, and protocols, such as the “Organic Law 2/2006 of May 3, on Education”, which establishes the foundations for preventing and intervening in cases of bullying [32].

In Italy, the fight against bullying is addressed through a combination of national, regional, and local policies and laws, including Italy’s Law No. 71, enacted on May 29, 2017. These action protocols encompass prevention and awareness measures, with a focus on promoting respect, empathy, and a zero-tolerance approach to bullying. Procedures for reporting incidents and providing psychological and emotional support to victims are in place. Disciplinary measures define sanctions and consequences for perpetrators. Additionally, there is an emerging trend of providing training to teachers and school staff on bullying-related materials for effective intervention. Parents play an active role in prevention and case resolution [33].

From a global perspective, the primary contribution of this study lies in providing insights into how bullying is distributed between two countries, Spain and Italy, as well as among different age groups and genders. These data can serve as a foundation for developing more precise and specific interventions aimed at combating bullying effectively.

The study’s first objective was to determine the differences in the prevalence of bullying in primary and secondary schools between Spain and Italy. 

The results reveal significant differences in the categorization of students based on the types of school bullying and their country of origin. In the Spanish sample, these differences are particularly evident in the categories of “no victim aggress”, “victim”, and “victim and aggress”, with a higher percentage of students in the roles of “victim and aggress” and “victim” compared to the Italian sample. Conversely, in the Italian sample, significant differences are observed for the same categories, with a higher percentage of students in the role of “no victim aggress” compared to the Spanish sample.

Consequently, these significant findings lead to the conclusion that residing in Spain or Italy does indeed impact the categorization of bullying. It becomes apparent that being Spanish increases the proportion of students categorized as “victim and aggress” and “victim” while decreasing the representation in the “no victim aggress” category. Conversely, being Italian is associated with an increase in the “no victim aggress” category and a decrease in “victim” and “victim and aggress”.

From an educational perspective, school bullying predominantly occurs among children and adolescents aged between 7 and 16, encompassing both the roles of the “aggress” and the “victim” [11]. While several European countries, such as Scadinavian nations, have been researching and addressing bullying since the 1970s, Italy began studying this phenomenon later, in the late 1990s, through research conducted by the Department of Psychology at the University of Florence. Since then, numerous scientific publications and educational initiatives have focused on this topic within specific schools and local contexts [11].

The initial investigation into violent behaviors in schools was conducted through questionnaires directed at students and teachers. Among the subjects who responded to these questionnaires, from December 2007 to April 2008, there were 5418 students and 592 teachers from secondary schools. The majority of them came from southern Italy and the surrounding islands. This study concluded that more than half of the students (51%) and over a third of the teachers (36%) had witnessed episodes of violence in school. Additionally, 37% of the students, also more than a third, reported personally experiencing school bullying by their peers [34].

However, upon additionally incorporating institutional data, which pertain to the values of the working group, it becomes evident that school bullying in Italy continues to be a widely spread phenomenon, characterized by territorial peculiarities and a higher frequency than that recorded in neighboring countries or areas with similar sociodemographic characteristics. Furthermore, it has been observed that only a few studies (n = 13) worldwide have accurately examined the prevalence of school bullying in the last decade, despite it being considered a significant global public health issue. This current study contributed to a precise assessment of the actual prevalence of bullying among school-age children in one of Italy’s most populous cities [34].

Similarly, Spain has also reported a significant increase in school bullying in recent years, now recognized as an active social problem [35]. However, unlike other European countries, Spain lacks a single program for intervention; instead, numerous plans are based on the autonomous communities and individual schools, which may negatively impact this phenomenon [36], as evidenced by our results. Additionally, teacher training is an essential aspect that has yet to be adequately addressed [37].

The second objective aimed to examine the differences in gender based on the country of origin and age among students regarding their experiences or involvement in school bullying.

The results revealed significant differences between gender and country concerning bullying categories. Specifically, these differences are observable in the categories of “no victim aggress” (14.4%) and “victim and aggress” (30.5%) in boys in Spain. Similarly, significant differences were noted in the “victim” category (25.0%) among girls in Spain. In the Italian sample, significant differences were found in the categories of “no victim aggress” (28.4%) and “victim” (16.8%) among Italian boys, as well as in the category of “no victim aggress” (40.3%) among Italian girls. No significant differences were observed for the remaining categories in all groups.

It is worth noting that the distribution of percentages by gender and country in relation to bullying categories is the same for Italian boys and girls and for Spanish boys and girls. In each of these groups, the category of “no victim aggress” has the highest percentage of students, followed by “victim”, “victim and aggress”, and, finally, “aggress”.

Therefore, the significant results from the comparison between gender and country lead to the conclusion that the combination of these variables does indeed influence the categorization of bullying. In this way, it can be observed that being a Spanish boy increases the proportion of “victim” and “victim and aggress” categories and decreases that of “no victim aggress”, while being a Spanish girl increases the proportion of being a “victim”. Conversely, being an Italian boy reduces the proportion of being a “victim” and increases the “no victim aggress” category, similar to being an Italian girl, while also decreasing the “victim and aggress” category.

Other studies also find a higher prevalence of aggressors and “victim and aggress” roles among males, while simultaneously observing a higher prevalence of “victims” and “no victim-aggress” roles among females [12,18,27,38,39]. In contrast, the results of the study by Górriz et al. [21] show significant findings with a higher presence of males in the victim role and females in the aggressor role. Thus, other research suggests that boys and male adolescents are more involved in the roles of “victims” and “no victim-aggress”, which does not align with the results of this study, where a higher percentage of girls are observed in these categories compared to boys [40].

The types of behaviors associated with school bullying also differ by gender. Physical violence, insults, or threats are more common among boys, while girls are associated with relational behaviors such as exclusion, spreading rumors, or being ignored by their peers [12,41,42]. The figures found within these gender and behavior categories can be explained by considering gender socialization and associated normative expectations, as school bullying can be understood as a behavior in which different genders act in accordance with what is expected of them [12,41,43].

These results suggest that both boys and girls internalize social stereotypes. For instance, the stereotype associated with masculinity, which includes traits of virility and violence, contrasts with the stereotype of femininity. These stereotypes are assimilated from an early age. Therefore, it would be logical for intervention strategies against bullying to be directed at challenging and dismantling these deeply rooted sexist stereotypes in society [44].

Regarding age and school bullying, significant differences were also found, with students with a lower average age being in the “victim” category, followed by the “no victim-aggress” category as age increases. Those with a higher average age are in the “agress” category. Likewise, significant differences were obtained between the “no victim-agress” category, the “victim” category, and the “agress” category based on age. Furthermore, significant differences were also observed between the “victim” category and students in the “agress and victim and agress” categories. Lastly, significant differences were observed based on the age of the students for the “agress” category compared to those in the “victim and agress” category.

In contrast to this research, other studies did not find significant differences in the various categories with respect to students’ age [21,40]. Some studies show higher results in the aggressor role between the ages of 11 and 15, as well as in another study where it was noted that students with higher involvement are in the middle grades of secondary school, decreasing in the higher grades [27].

As emphasized in other research, there is an urgent need to establish interventions to prevent both victimization and aggression in school bullying among school-age students [45]. Evidence shows that the most effective antibullying interventions are those that emphasize violence prevention and, even more so, promote positive coexistence and a school culture based on respect and good treatment. However, educational institutions often develop strategies to address bullying only when it is already present in the institution, which means that these measures are reactive rather than proactive [46].

Lastly, with regard to the research’s limitations, it is important to acknowledge that the findings may not be universally applicable to all children and adolescents in Spain, Italy, or other regions, which could potentially undermine their external validity. Additionally, the use of self-report questionnaires like the EBIPQ presents a potential limitation in research, as it necessitates cautious interpretation. Despite this, it is worth noting that the EBIPQ is a questionnaire with robust psychometric properties and validation. A key takeaway from this research is the imperative of continuing to explore interventions aimed at mitigating this issue. Furthermore, the significant variations observed across countries, as well as by gender and age, offer valuable insights that can inform the development of more suitable psychoeducational intervention objectives.

The study on bullying in the countries of Spain and Italy yields significant practical implications in the field of education and society at large. Some of the key conclusions include the need for awareness and prevention; both countries should prioritize raising awareness about bullying and implementing prevention programs in schools. This would help reduce the incidence of bullying cases and foster a safe and healthy school environment. Additionally, the gender differences identified in the study require specific attention, underscoring the need to address gender disparities in prevention and support strategies while promoting gender equality in education. Early detection and intervention are also essential; educational systems and healthcare professionals need to be trained to effectively identify and address bullying, offering support to both victims and aggressors. Thus, this study emphasizes the importance of addressing this significant issue from a multidisciplinary and gender perspective, aiming to create a safe and respectful educational environment for all students. Awareness, prevention, and support for victims are fundamental steps towards eradicating bullying in these societies.

## 5. Conclusions

Statistically significant differences were found in the categorization of subjects in terms of bullying types and country, indicating that there were more Italians than Spaniards in the “no victim aggress” category and more Spaniards in the roles of “victim” and “victim and aggress”.

Similarly, statistically significant differences were found between bullying categories and gender combined with the country of origin. Differences were observed for Spanish boys in the categories “victim and aggress” and “no victim aggress” and for Spanish girls in the “victim” category. Regarding Italian boys, differences were found in the “aggress” and “victim and aggress” categories and for Italian girls in the “no victim aggress” and “victim and aggress” categories. It is noteworthy that all of them had a higher percentage of students in the “no victim aggress” category, followed by “victim”, “victim and aggress”, and, finally, “aggress”.

Significant results were also obtained between students’ age and bullying categories, with students with the lowest mean age belonging to the “victim” category, followed by “no victim aggress”, “victim and aggress”, and, finally, those with the highest mean age in the “aggress” category.

The statistically significant differences identified in the research, involving the students’ country of origin (Spain vs. Italy), along with gender and age in relation to school bullying, provide contemporary insights that could be valuable in formulating unified approaches across countries. These interventions are designed to curb the continuation of school bullying, which can have consequences in education, including a negative impact on academic performance and an increased risk of students dropping out. Additionally, it can have adverse effects on the mental health and overall quality of life for children and adolescents.

In summary, these findings underscore the significance of addressing school bullying through a gender-oriented lens and tailoring strategies to accommodate the distinctive characteristics of each educational stage. The implementation of prevention measures and fostering of safe and respectful school environments for all students, regardless of the roles they may occupy in bullying dynamics, are essential aspects of this effort.

## Figures and Tables

**Table 1 children-10-01762-t001:** Statistical analysis using chi-squared between the bullying categories and the country.

Country of Origin of the Sample		Bullying Category				Total
		No Victim Aggress	Victim	Aggress	Victim and Aggress	
Spain	Count	322	139	20	81	562
	Expected	376.9	102.4	19	63.7	562
	% within Country	57.3%	24.7%	3.6%	14.4%	100.0%
	% within Bullying category	31.3%	49.6%	38.5%	46.6%	36.6%
	% of Total	21.0%	9.0%	1.3%	5.3%	36.6%
	Adjusted Residual	−6.2 _a_	5.0 _b_	0.3	2.9 _c_	
Italy	Count	708	141	32	93	974
	Expected	653.1	177.6	33.0	110.3	974
	% within Country	72.7%	14.5%	3.3%	9.5%	100.0%
	% within Bullying category	68.7%	50.4%	61.5%	53.4%	63.4%
	% of Total	46.1%	9.2%	2.1%	6.1%	63.4%
	Adjusted Residual	6.2 _a_	−5.0 _b_	−0.3	−2.9 _c_	
Total	Count	1030	280	52	174	1536
	Expected	1030	280	52	174	1536
	% within Country	67.1%	18.2%	3.4%	11.3%	100.0%
	% within Bullying category	100.0%	100.0%	100.0%	100.0%	100.0%
	% of Total	67.1%	18.2%	3.4%	11.3%	100.0%

_a_. Significant differences for the “no victim aggress” category; _b_. significant differences for the “victim” category; _c_. significant differences for the “victim and aggress” category.

**Table 2 children-10-01762-t002:** Statistical analysis using chi-squared between the bullying categories, country, and gender.

Gender		Bullying Category				Total
		No Victim Aggress	Victim	Aggress	Victim and Aggress	
Boys Spain	Count	148	69	14	53	284
	Expected	190.4	51.8	9.6	32.2	284.0
	% within Gender and country	52.1%	24.3%	4.9%	18.7%	100.0%
	% within Bullying category	14.4%	24.6%	26.9%	30.5%	18.5%
	% of Total	9.6%	4.5%	0.9%	3.5%	18.5%
	Adjusted Residual	−5.9 _a_	2.9 _b_	1.6	4.3 _c_	
Girls Spain	Count	174	70	6	28	278
	Expected	186.4	50.7	9.4	31.5	278.0
	% within Gender and country	62.6%	25.2%	2.2%	10.1%	100.0%
	% within Bullying category	16.9%	25.0%	11.5%	16.1%	18.1%
	% of Total	11.3%	4.6%	0.4%	1.8%	18.1%
	Adjusted Residual	−1.8	3.3 _b_	−1.3	−0.7	
Boys Italy	Count	293	47	13	46	399
	Expected	267.6	72.7	13.5	45.2	399.0
	% within Gender and country	73.4%	11.8%	3.3%	11.5%	100.0%
	% within Bullying category	28.4%	16.8%	25.0%	26.4%	26.0%
	% of Total	19.1%	3.1%	0.8%	3.0%	26.0%
Girls Italy	Adjusted Residual	3.1 _a_	−3.9 _b_	−0.2	0.1	
	Count	415	94	19	47	575
	Expected	385.6	104.8	19.5	65.1	575.0
	% within Gender and country	72.2%	16.3%	3.3%	8.2%	100.0%
	% within Bullying category	40.3%	33.6%	36.5%	27.0%	37.4%
	% of Total	27.0%	6.1%	1.2%	3.1%	37.4%
	Adjusted Residual	3.3 _a_	−1.5	−0.1	−3.0 _c_	
Total	Count	1030	280	52	174	1536
	% within Gender and country	67.1%	18.2%	3.4%	11.3%	100.0%
	% within Bullying category	100.0%	100.0%	100.0%	100.0%	100.0%
	% of Total	67.1%	18.2%	3.4%	11.3%	100.0%

_a_. Significant differences for the “no victim aggress” category; _b._ significant differences for the “victim” category; _c._ significant differences for the “victim and aggress” category.

**Table 3 children-10-01762-t003:** Descriptive analysis of an ANOVA test between the bullying categories and age.

Age								
	N	Mean	Std. Deviation	Std. Error	95% Confidence Interval for Mean		Min	Max
					Lower Bound	Upper Bound		
No victim aggress	1030	13.05	2.229	0.069	12.91	13.18	10	19
Victim	280	12.54	1.883	0.113	12.32	12.76	10	18
Aggress	52	13.92	2.204	0.306	13.31	14.54	10	18
Victim and aggress	174	13.25	2.279	0.173	12.91	13.59	10	19
Total	1536	13.01	2.190	0.056	12.90	13.12	10	19

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
