# Peer review of "Intercultural Differences between Spain and Italy Regarding School Bullying, Gender, and Age"

_children, 2023, doi:10.3390/children10111762_

Round 1

Reviewer 1 Report

Comments and Suggestions for Authors

Thank you for the opportunity to review this study entitled “Intercultural Differences Between Spain and Italy Regarding School Bullying, Gender, and Age (children-2646874).

The paper presents an exploration of the differences between Spain and Italy regarding the presence of bullying in primary and secondary schools, also considering the role of Gender and Age. A sample of 1372 students (from Italy and Spain) was involved in the research.

In my opinion, the research topic is relevant. However, some issues need to be addressed before the paper is ready for publication.

·       Abstract “The objectives of this research were to establish the relationship between Spain and Italy 13 regarding the presence of bullying in primary and secondary schools”. In my opinion, the word “relationship” is not correct, since the authors assessed differences. Please reformulate both in the abstract and in the whole manuscript.

·       Abstract line 22: a full stop is missing.

·       More information about the European Bullying Intervention Project Questionnaire (EBIPQ) should be provided (e.g., internal consistency in the present sample).

·       Procedures: “Participants from different classrooms signed informed consent forms to participate”. The participants? So did the minors sign? Or whoever has their custody (e.g., parents)? Please specify.

·       “To study the correlation between the different categories related to bullying obtained with the EBIPQ scale and age, a one-way ANOVA was conducted, along with post-hoc testing”. The one-way ANOVA did not assess correlation. Please correct.

·       Conclusions: Please explain the practical implications of this research in a broad and detailed manner.

Best wishes

Author Response

Mrs. Jessica Fernández Solana

Department of health sciences

University of Burgos, Paseo Comendadores s/n.

Burgos, 09001, Spain

Tel. (+34) 947499108

19-10-2023

Children.  Subject: Submissions Needing Revision

Dear editor.

Thank you very much for inviting us to submit our response to reviewers for our manuscript (children-2646874) entitled: “Intercultural Differences Between Spain and Italy Regarding School Bullying, Gender, and Age”

We have checked our manuscript according to the Academic Editor, the reviewers’ comments and the Journal requirements. We have also responded to some comments from reviewers point by point).

We would be very grateful if you could consider our manuscript to be published in your journal.

Yours sincerely,

Jessica Fernández Solana, OT, PT

  1. Response to Reviewer 1:

First of all, we would like to express our sincere gratitude for all comments and suggestions received from the Reviewer 1. This information has certainly enriched the text for its best understanding, thank you very much indeed. We have clarified the reviewer1’s questions. We have introduced the required changes both in our answers to the specific comments and in the final manuscript V2.

Thank you for the opportunity to review this study entitled “Intercultural Differences Between Spain and Italy Regarding School Bullying, Gender, and Age” (children-2646874).

The paper presents an exploration of the differences between Spain and Italy regarding the presence of bullying in primary and secondary schools, also considering the role of Gender and Age. A sample of 1372 students (from Italy and Spain) was involved in the research.

Response: The total number of the sample has been modified due to an error in the sum, the total number of the sample equals 1536.

In my opinion, the research topic is relevant. However, some issues need to be addressed before the paper is ready for publication.

Response: Thank you very much for your suggestions, we will try to respond accurately and attentively to each of your contributions.

  • Abstract “The objectives of this research were to establish the relationship between Spain and Italy 13 regarding the presence of bullying in primary and secondary schools”. In my opinion, the word “relationship” is not correct, since the authors assessed differences. Please reformulate both in the abstract and in the whole manuscript.

Response: Thank you very much for your input, we have modified the objective of the study in response to your suggestion.

The aim of the study is: To establish significant differences between a sample of students from Spain and Italy regarding the presence of bullying in primary and secondary schools".

  • Abstract line 22: a full stop is missing.

Response: has been added in the manuscript.

  • More information about the European Bullying Intervention Project Questionnaire (EBIPQ) should be provided (e.g., internal consistency in the present sample).

Response: In the instrument description section, the internal consistency of the instrument for the sample analysed has been added (see lines 124-125). This was 0.852.

  • Procedures: “Participants from different classrooms signed informed consent forms to participate”. The participants? So did the minors sign? Or whoever has their custody (e.g., parents)? Please specify.

Response: Thank you for your input. As the participants were minors, they were unable to sign the informed consent form. The informed consents, after contacting the schools and having them agree to give us permission to collect data in the respective schools, the parents or legal guardians signed the consent for data collection. The following information has been added to the lines 135-137.

  • “To study the correlation between the different categories related to bullying obtained with the EBIPQ scale and age, a one-way ANOVA was conducted, along with post-hoc testing”. The one-way ANOVA did not assess correlation. Please correct.

Response: Thank you for your input, the text has been modified on the line 149.

  • Conclusions: Please explain the practical implications of this research in a broad and detailed manner.

Response: Thank you for your proposal. Some practical implications have been added in the results section, which in a detailed way can add quality to the manuscript, as well as further explanation for the reader or for future research (see lines 362-375).

 Best wishes

We hope we have now answered all your comments and we are looking forward to hearing from you again.

Jessica Fernández Solana, OT, PT

Reviewer 2 Report

Comments and Suggestions for Authors

 The sample used is very important and the results very significant.  The authors can improve their article:

1. Define more precisely the characteristics of the sample:

1.1. Expand the information on the sample:

- Delimit the geographic area where the centers are located, their rural or urban enclave, size of the centers, dependence on the administration, etc. 

In other words, information that makes it possible to establish whether or not the characteristics of the samples are homogeneous between the two countries. And relate all this to previous studies discussed in lines 242-243.

2. The authors should present the existing official protocols on the actions that the autonomous governments or associations have developed in each country, and indicate how to implement them.

3. The conclusions should be improved in relation to the objectives presented in the abstract. 

Author Response

Mrs. Jessica Fernández Solana

Department of health sciences

University of Burgos, Paseo Comendadores s/n.

Burgos, 09001, Spain

Tel. (+34) 947499108

19-10-2023

Children.  Subject: Submissions Needing Revision

Dear editor.

Thank you very much for inviting us to submit our response to reviewers for our manuscript (children-2646874) entitled: “Intercultural Differences Between Spain and Italy Regarding School Bullying, Gender, and Age”

We have checked our manuscript according to the Academic Editor, the reviewers’ comments and the Journal requirements. We have also responded to some comments from reviewers point by point).

We would be very grateful if you could consider our manuscript to be published in your journal.

Yours sincerely,

Jessica Fernández Solana, OT, PT

  1. Response to Reviewer 2:

First of all, we would like to express our sincere gratitude for all comments and suggestions received from the Reviewer 2. This information has certainly enriched the text for its best understanding, thank you very much indeed. We have clarified the reviewer2’s questions. We have introduced the required changes both in our answers to the specific comments and in the final manuscript V2.

The sample used is very important and the results very significant.  The authors can improve their article:

Response: Thank you very much for your contributions, we will try to respond to each of them in a detailed and careful way.

  1. Define more precisely the characteristics of the sample:

1.1. Expand the information on the sample:

- Delimit the geographic area where the centers are located, their rural or urban enclave, size of the centers, dependence on the administration, etc.

In other words, information that makes it possible to establish whether or not the characteristics of the samples are homogeneous between the two countries. And relate all this to previous studies discussed in lines 242-243.

Response: Thank you for your comment. Some of the data we have are included in section 2.1 Participants, where we have included the administration unit, the number of students per year and the geographical area to which the schools belong, in both countries in an urban environment.

  1. The authors should present the existing official protocols on the actions that the autonomous governments or associations have developed in each country, and indicate how to implement them.

Response: Thank you very much for your input, the following information has been included between lines 231 and 249.

  1. The conclusions should be improved in relation to the objectives presented in the abstract.

Response: Thank you for your comment, efforts have been made to improve the wording of the conclusions and adapt them to the objectives presented.

We hope we have now answered all your comments and we are looking forward to hearing from you again.

Jessica Fernández Solana, OT, PT

Reviewer 3 Report

Comments and Suggestions for Authors

Congratulations to the authors for the work.

Some improvements are suggested

It would be necessary to include more data on the questionnaire. 

In the abstract they speak of a sample of 1372 and in methodology and results of 562 Spaniards and 974 Italians.

First of all, how long before the survey was asked about the occurrence of bullying? How was the translation into Italian? What was the reliability of the scale? How are the roles of victims, aggressors and aggressors determined? ....? It would be useful to indicate the gender distinction of participants from each country and the average age.

The discussion should report the most significant results, such as percentages of occurrence in both countries.

The conclusion should be more clarifying. The results should be discussed in comparison with other studies taking into account the time frame of the questionnaire, for example there are instruments that ask for the previous two months and others for the occurrence in the previous year, this determines the occurrence. Likewise, to determine the roles, there may be studies that consider the occurrence if the behavior occurs once and others if it occurs repeatedly over time according to the definition of bullying.

Many quotes are used in the discussion that have not been considered in the introduction.

Author Response

Mrs. Jessica Fernández Solana

Department of health sciences

University of Burgos, Paseo Comendadores s/n.

Burgos, 09001, Spain

Tel. (+34) 947499108

19-10-2023

Children.  Subject: Submissions Needing Revision

Dear editor.

Thank you very much for inviting us to submit our response to reviewers for our manuscript (children-2646874) entitled: “Intercultural Differences Between Spain and Italy Regarding School Bullying, Gender, and Age”

We have checked our manuscript according to the Academic Editor, the reviewers’ comments and the Journal requirements. We have also responded to some comments from reviewers point by point).

We would be very grateful if you could consider our manuscript to be published in your journal.

Yours sincerely,

Jessica Fernández Solana, OT, PT

  1. Response to Reviewer 3:

First of all, we would like to express our sincere gratitude for all comments and suggestions received from the Reviewer 3. This information has certainly enriched the text for its best understanding, thank you very much indeed. We have clarified the reviewer3’s questions. We have introduced the required changes both in our answers to the specific comments and in the final manuscript V2.

Congratulations to the authors for the work.

Some improvements are suggested

Response: Thank you very much for your input and for the review work done. We will try to respond in a detailed and careful way to each of your contributions.

It would be necessary to include more data on the questionnaire.

Response: Thank you very much for your comment. An attempt has been made to include more information on the questionnaire used in the material and methods section.

In the abstract they speak of a sample of 1372 and in methodology and results of 562 Spaniards and 974 Italians.

Response: Thank you for your comment. The number of the overall sample has been modified in methodology and results. The correct number, adding the sample collected in Spain and Italy, is 1536.

First of all, how long before the survey was asked about the occurrence of bullying? How was the translation into Italian? What was the reliability of the scale? How are the roles of victims, aggressors and aggressors determined? ....? It would be useful to indicate the gender distinction of participants from each country and the average age.

Response: Thank you for your comment, the following information has been added to the text. Information on the reliability of the questionnaire used has been added in the material and methods section. The roles of bullying are determined by the questionnaire itself through summations of the items that determine victimhood or aggression; these 2 summations are recoded and at the same time a summation is done again with the same items, in such a way that

the four categories of bullying are obtained. After completion, depending on the score obtained and using the guidelines for interpretation, we know to which group each participant belongs.

Also, in the results section, more information about the socio-demographic characteristics of the sample has been added, indicating in more detail the characteristics for the Spanish and Italian sample (see lines 159-162).

The discussion should report the most significant results, such as percentages of occurrence in both countries.

Response: Thank you for your comment. The most significant data found in the research, as well as the percentages of buylling occurrence for each country, have been added to the discussion.

The conclusion should be more clarifying. The results should be discussed in comparison with other studies taking into account the time frame of the questionnaire, for example there are instruments that ask for the previous two months and others for the occurrence in the previous year, this determines the occurrence. Likewise, to determine the roles, there may be studies that consider the occurrence if the behavior occurs once and others if it occurs repeatedly over time according to the definition of bullying.

Response: In response to your comment, the conclusion has been modified to explain in more detail the results found in our study and in line with our objective.

Many quotes are used in the discussion that have not been considered in the introduction.

Response: Thank you for your comment. Some of the quotations used in the discussion have been added in the introduction section to improve the quality of the manuscript.

We hope we have now answered all your comments and we are looking forward to hearing from you again.

Jessica Fernández Solana, OT, PT

Reviewer 4 Report

Comments and Suggestions for Authors

This is an interesting study but I believe seriously confounded at present. It is possible that a reanalysis of the data will make this study contribute significantly to the literature. Below you will find my specific concerns.

1. My biggest concern is with the inequality of the samples across key demographic variables. For example, males in the Italian sample are at 40% but in the Spanish sample at 51%. Thus, any subsequent differences could be attributed to the well known differences between males and females rather than actual between country differences. This serious methodological concern can be corrected if the authors utilize  propensity score matching or any matching procedure to create equivalent samples across countries on gender and age groups. If they do that, then they can reanalyze the data and report on true between countries differences. 

2. If the authors decide to revise, they should also address the following issues: (a) how classification into bullies and victims was conducted in light  of the used instrument (b) provide the instrument in the appendix, (c) indicate the scaling of the instrument, (d) report on the reliability and validity of the instrument, regardless of past findings but with the current samples, (e) conduct measurement invariance of the bullying scale across countries and across gender, (f) provide values for the statistical tests and their respective degrees of freedom rather than only p-values.

I hope this information is useful as the authors contemplate revising their work.

Comments on the Quality of English Language

Some parts especially in results were difficult to understand.

Author Response

Mrs. Jessica Fernández Solana

Department of health sciences

University of Burgos, Paseo Comendadores s/n.

Burgos, 09001, Spain

Tel. (+34) 947499108

19-10-2023

Children.  Subject: Submissions Needing Revision

Dear editor.

Thank you very much for inviting us to submit our response to reviewers for our manuscript (children-2646874) entitled: “Intercultural Differences Between Spain and Italy Regarding School Bullying, Gender, and Age”

We have checked our manuscript according to the Academic Editor, the reviewers’ comments and the Journal requirements. We have also responded to some comments from reviewers point by point).

We would be very grateful if you could consider our manuscript to be published in your journal.

Yours sincerely,

Jessica Fernández Solana, OT, PT

  1. Response to Reviewer 4:

First of all, we would like to express our sincere gratitude for all comments and suggestions received from the Reviewer 4. This information has certainly enriched the text for its best understanding, thank you very much indeed. We have clarified the reviewer4’s questions. We have introduced the required changes both in our answers to the specific comments and in the final manuscript V2.

This is an interesting study but I believe seriously confounded at present. It is possible that a reanalysis of the data will make this study contribute significantly to the literature. Below you will find my specific concerns.

Response: Thank you for your comments, we will try to address them all independently in a specific, careful and detailed manner.

  1. My biggest concern is with the inequality of the samples across key demographic variables. For example, males in the Italian sample are at 40% but in the Spanish sample at 51%. Thus, any subsequent differences could be attributed to the well known differences between males and females rather than actual between country differences. This serious methodological concern can be corrected if the authors utilize propensity score matching or any matching procedure to create equivalent samples across countries on gender and age groups. If they do that, then they can reanalyze the data and report on true between countries differences.

Response: Thank you very much for your input, you are right so the statistic has been redone regarding the differences in bullying categories for gender. In order to see the real differences between countries regarding gender, 4 categories have been created (boy Spain/girl Spain and boy Italy/girl Italy). Therefore table 2 has been modified and the results have been modified.

  1. If the authors decide to revise, they should also address the following issues: (a) how classification into bullies and victims was conducted in light of the used instrument (b) provide the instrument in the appendix, (c) indicate the scaling of the instrument, (d) report on the reliability and validity of the instrument, regardless of past findings but with the current samples, (e) conduct measurement invariance of the bullying scale across countries and across gender, (f) provide values for the statistical tests and their respective degrees of freedom rather than only p-values.

Response: The instrument has been added in the appendix. In addition, data on the reliability and validity of the instrument for the current samples have been added in section 2.2 Instrument on the line 124-125. Values for statistical tests have been added in the results section (see line 166, 180 and 209).

I hope this information is useful as the authors contemplate revising their work.

Comments on the Quality of English Language

Some parts especially in results were difficult to understand.

Response: Thank you for your input. We have revised the language throughout the manuscript and tried to clarify the results.

We hope we have now answered all your comments and we are looking forward to hearing from you again.

Jessica Fernández Solana, OT, PT

Round 2

Reviewer 3 Report

Comments and Suggestions for Authors

Congratulations to the authors for the corrections.

Some small improvements are recommended.

It would be convenient to carry out the analysis of the internal consistency of the questionnaire on the one hand for the victimisation variable and on the other hand for the aggression variable.

It is not clear how the roles of victimiser, aggressor, aggressor-victimised and not involved are established. See for example reference (28) page 165 "Para establecer losdiferentes roles.... en al menos uno de lo sítems de agresión y devictimización."

No sense is made of the statements between lines 119-120, as there are no distinctions in the study of type of direct, indirect, verbal, physical or social behaviour, nor do these references (27 and 28) make such distinctions.

Author Response

Mrs. Jessica Fernández Solana

Department of health sciences

University of Burgos, Paseo Comendadores s/n.

Burgos, 09001, Spain

Tel. (+34) 947499108

25-10-2023

Children.  Subject: Submissions Needing Revision

Dear editor.

Thank you very much for inviting us to submit our response to reviewers for our manuscript (children-2646874) entitled: “Intercultural Differences Between Spain and Italy Regarding School Bullying, Gender, and Age”

We have checked our manuscript according to the Academic Editor, the reviewers’ comments and the Journal requirements. We have also responded to some comments from reviewers point by point).

We would be very grateful if you could consider our manuscript to be published in your journal.

Yours sincerely,

Jessica Fernández Solana, OT, PT

  1. Response to Reviewer 3:

First of all, we would like to express our sincere gratitude for all comments and suggestions received from the Reviewer 3. This information has certainly enriched the text for its best understanding, thank you very much indeed. We have clarified the reviewer3’s questions. We have introduced the required changes both in our answers to the specific comments and in the final manuscript V3.

Congratulations to the authors for the corrections.

Some small improvements are recommended.

It would be convenient to carry out the analysis of the internal consistency of the questionnaire on the one hand for the victimisation variable and on the other hand for the aggression variable.

Response: Thank you for your comment, the analysis of the internal consistency of the aggression variable and the victimisation variable has been carried out separately. This information has been included in lines 140-141.

“Additionally, the internal consistency analysis for the first 7 victimization items is 0.796, and for the following 7 items, it is 0.804.”

It is not clear how the roles of victimiser, aggressor, aggressor-victimised and not involved are established. See for example reference (28) page 165 "Para establecer losdiferentes roles.... en al menos uno de lo sítems de agresión y devictimización."

Response: thank you for your comment. We have used the indications of the scale for the calculation of the scores and the establishment of the 4 roles within bullying.

Thank you for your contribution, I found it very interesting and we have included in the description of the instrument the scoring procedure to establish the roles (see lines 128-138).

No sense is made of the statements between lines 119-120, as there are no distinctions in the study of type of direct, indirect, verbal, physical or social behaviour, nor do these references (27 and 28) make such distinctions.

Response: is right that we have referred to a part that has not been taken as a reference for this study.

We hope we have now answered all your comments and we are looking forward to hearing from you again.

Jessica Fernández Solana, OT, PT

Reviewer 4 Report

Comments and Suggestions for Authors

Thank you for your efforts to accommodate my earlier comments. The revised version is undoubtedly improved but not yet where it should be. I believe there are a few important items to address still.

1. There was no information about reliability and validity of the measure or measurement invariance across countries. In the absence of that information we cant know if the construct measured is the same across countries.

2. Tables and text that indicate significant tests must include the statistical tests not just p-values. Especially the tables must have letter subscripts indicating which groups are different from each other.

Comments on the Quality of English Language

English proofreading is necessary.

Author Response

Mrs. Jessica Fernández Solana

Department of health sciences

University of Burgos, Paseo Comendadores s/n.

Burgos, 09001, Spain

Tel. (+34) 947499108

25-10-2023

Children.  Subject: Submissions Needing Revision

Dear editor.

Thank you very much for inviting us to submit our response to reviewers for our manuscript (children-2646874) entitled: “Intercultural Differences Between Spain and Italy Regarding School Bullying, Gender, and Age”

We have checked our manuscript according to the Academic Editor, the reviewers’ comments and the Journal requirements. We have also responded to some comments from reviewers point by point).

We would be very grateful if you could consider our manuscript to be published in your journal.

Yours sincerely,

Jessica Fernández Solana, OT, PT

  1. Response to Reviewer 4:

First of all, we would like to express our sincere gratitude for all comments and suggestions received from the Reviewer 4. This information has certainly enriched the text for its best understanding, thank you very much indeed. We have clarified the reviewer4’s questions. We have introduced the required changes both in our answers to the specific comments and in the final manuscript V2.

Thank you for your efforts to accommodate my earlier comments. The revised version is undoubtedly improved but not yet where it should be. I believe there are a few important items to address still.

  1. There was no information about reliability and validity of the measure or measurement invariance across countries. In the absence of that information we cant know if the construct measured is the same across countries.

Response: Gracias por su comentario. Ha sido añadido el análisis de consistencia interna de la escala en ambas muestras, tanto Española como italana (see lines 141-142). En este caso es de 0.852 para la muestra española y 0.850 para la italiana.

  1. Tables and text that indicate significant tests must include the statistical tests not just p-values. Especially the tables must have letter subscripts indicating which groups are different from each other.

Response: Thank you very much for your comment, although we do not quite understand what you mean. We have changed the names of the tables indicating the statistical test performed in each of them. Likewise, for the first 2 tables, although the value of the corrected residual is included, a letter in subscript has been added next to it to indicate the significant differences per group.

English proofreading is necessary.

Response: The entire manuscript has been reviewed and revised in English and modifications have been made.

We hope we have now answered all your comments and we are looking forward to hearing from you again.

Jessica Fernández Solana, OT, PT
